

# A new species of *Gulo* from the Early Pliocene Gray Fossil Site (Eastern United States); rethinking the evolution of wolverines

Joshua X. Samuels[1,2], Keila E. Bredehoeft[2] and Steven C. Wallace[1,2]

[1] Department of Geosciences, East Tennessee State University, Johnson City, TN, United States of America
[2] Don Sundquist Center of Excellence in Paleontology, East Tennessee State University, Johnson City, TN, United States of America

## ABSTRACT

The wolverine (*Gulo gulo*) is the largest living terrestrial member of the Mustelidae; a versatile predator formerly distributed throughout boreal regions of North America and Eurasia. Though commonly recovered from Pleistocene sites across their range, pre-Pleistocene records of the genus are exceedingly rare. Here, we describe a new species of *Gulo* from the Gray Fossil Site in Tennessee. Based on biostratigraphy, a revised estimate of the age of the Gray Fossil Site is Early Pliocene, near the Hemphillian—Blancan transition, between 4.9 and 4.5 Ma. This represents the earliest known occurrence of a wolverine, more than one million years earlier than any other record. The new species of wolverine described here shares similarities with previously described species of *Gulo*, and with early fishers (*Pekania*). As the earliest records of both *Gulo* and *Pekania* are known from North America, this suggests the genus may have evolved in North America and dispersed to Eurasia later in the Pliocene. Both fauna and flora at the Gray Fossil Site are characteristic of warm/humid climates, which suggests wolverines may have become 'cold-adapted' relatively recently. Finally, detailed comparison indicates *Plesiogulo*, which has often been suggested to be ancestral to *Gulo*, is not likely closely related to gulonines, and instead may represent convergence on a similar niche.

# INTRODUCTION

Wolverines (*Gulo gulo*) are the largest living terrestrial mustelid, have a circumboreal Holarctic historic distribution, and have commonly been considered a 'cold-adapted' species (*Bonifay, 1971*; *Pasitschniak-Arts & Larivière, 1995*; *Zigouris et al., 2013*). Their diet is best characterized as being that of an opportunistic hypercarnivore, either ambushing and/or chasing prey or scavenging carcasses (*Pasitschniak-Arts & Larivière, 1995*). Though primarily terrestrial, they are capable climbers, swimmers, and diggers (*Pasitschniak-Arts & Larivière, 1995*; and citations therein). Though wolverines are well-known, their fossil record is relatively sparse and their origin has been controversial (ex. *Kurtén, 1970*; *Harrison, 1981*).

Corresponding author
Joshua X. Samuels,
samuelsjx@etsu.edu

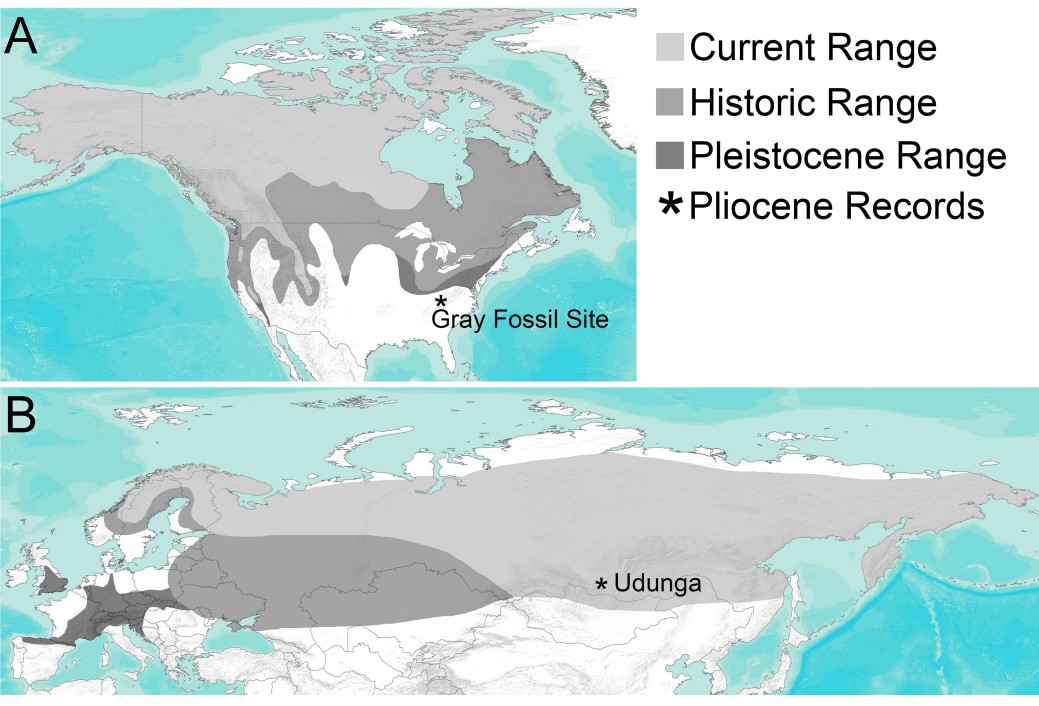

**Figure 1 Distribution of *Gulo* through time in (A) North America and (B) Eurasia.** Current range (light gray) and historic range (medium gray) are based on *Pasitschniak-Arts & Larivière (1995)*, *Zigouris et al. (2013)*; Pleistocene range (dark gray) is based on localities where fossils of *Gulo* have been found and reported in a wide range of literature sources. The only two known Pliocene occurrences, the Gray Fossil Site in Tennessee and Udunga Fauna of Russia, are highlighted with an asterisk. Terrain base map from ESRI ArcMap 10.5.

Fossil wolverines have been described from many Pleistocene and Holocene sites in North America and Eurasia, including quite a few outside of the species' historic distribution (Fig. 1, Table S1 and citations therein). *Bryant (1987)* compared a large sample of *Gulo* from the Pleistocene of the Yukon to extant populations and to other Pleistocene records of the genus from North America, which had been described as *Gulo gidleyi* (*Hall, 1936*) and later referred to *Gulo schlosseri* (*Kurtén & Anderson, 1980*). While *Bryant (1987)* found some small differences in average size, there was substantial overlap in size ranges and morphology between samples, leading him to conclude that all Quaternary *Gulo* should be treated as a single species. He did caution that a thorough study of Asian and European specimens was needed (*Bryant, 1987*), and several subsequent papers, largely focused on European material, have maintained *G. schlosseri* as a separate species (*Pasitschniak-Arts & Larivière, 1995*; *Kolfschoten, 2001*; *Nadachowski et al., 2011*; *Krajcarz, 2012*). Until a detailed analysis of specimens from all continents is undertaken, we will follow the taxonomy for Pleistocene wolverines from Eurasia in those studies. Specifically, following *Bryant (1987)*, all Quaternary records of *Gulo* in North America are considered *G. gulo* here.

The origin of wolverines has long been controversial (ex. *Zdansky, 1924*; *Viret, 1939*; *Kurtén, 1970*; *Hendey, 1978*; *Harrison, 1981*; *Alcalá, Montoya & Morales, 1994*; *Pasitschniak-Arts & Larivière, 1995*; *Montoya, Morales & Abella, 2011*); mostly due to the sparse pre-Pleistocene records. Only a single record of *Gulo* prior to the Quaternary has been reported; *Gulo minor* from the middle Pliocene age Udunga fauna from the Transbaikal region of Russia (*Sotnikova, 1982*; *Sotnikova, 1995*; *Erbajeva & Alexeeva, 2013*). *G. minor* is morphologically similar to, but substantially smaller than *G. gulo*. It falls below the range of size variation reported for large extant samples of *G. gulo* (*Bryant, 1987*; *Anderson, 1998*), as well as fossil samples from Europe (*Kormos, 1914*; *Bonifay, 1971*; *Kurtén, 1973*; *Döppes, 2001*) and North America (*Gidley & Gazin, 1938*; *Kurtén, 1973*; *Bryant, 1987*; *Anderson, 1998*). *Plesiogulo*, known from Middle to Late Miocene of Eurasia, Late Miocene and Early Pliocene of North America, and Late Miocene to Early Pliocene of Africa, has often been discussed as a potential close relative of *Gulo*. While some authors have considered *Plesiogulo* to be directly ancestral to *Gulo* (*Viret, 1939*; *Kurtén, 1970*; *Pasitschniak-Arts & Larivière, 1995*), others considered such a line of descent to be unlikely (*Zdansky, 1924*; *Webb, 1969*; *Hendey, 1978*; *Harrison, 1981*; *Xu & Wei, 1987*; *Alcalá, Montoya & Morales, 1994*; *Montoya, Morales & Abella, 2011*). Alternatively, some authors have suggested independent origins of *Gulo* and *Plesiogulo*, with *Gulo* originating from a fisher or marten-like ancestor in the late Miocene (*Samuels & Cavin, 2013*) and *Plesiogulo* from an early Miocene ischyrictine like *Iberictis* (*Ginsburg & Morales, 1992*; *Alcalá, Montoya & Morales, 1994*; *Montoya, Morales & Abella, 2011*).

Molecular data from a wide range of studies indicate *Gulo* is part of a clade (Guloninae) that includes fishers (*Pekania*), martens (*Martes*), and the tayra (*Eira*). Among gulonines, wolverines have been consistently found to be most closely related to martens, which form a sister group to the fisher (*Koepfli et al., 2008*; *Wolsan & Sato, 2010*; *Sato et al., 2012*; *Li et al., 2014*; *Malyarchuk, Derenko & Denisova, 2015*; *Zhu et al., 2016*). *Koepfli et al. (2008)* estimated the Guloninae diverged from other mustelids around 11.0 Ma (95% CI [9.4–12.5 Ma]); whereas *Sato et al. (2012)* estimated 12.65 Ma (95% CI [10.83–14.72 Ma]). *Li et al. (2014)* examined the complete mitochondrial genomes of gulonines and estimated the *Gulo-Martes* clade diverged from *Pekania* between 8.9 and 7.05 Ma, most likely around 7.6 Ma, while the divergence of *Gulo* and *Martes* was between 7.6 and 5.3 Ma, most likely around 6.4–6.3 Ma. Similarly, *Malyarchuk, Derenko & Denisova (2015)* estimated the divergence of *Gulo* and *Martes* to be around 5.6 Ma (95% CI [6.3–4.9] Ma). These estimates coincide with the earliest records of definite gulonine mustelids in the late Miocene, namely *Pekania occulta* from North America and *P. palaeosinensis* from Asia (*Wang, Tseng & Takeuchi, 2012*; *Samuels & Cavin, 2013*), and *Martes ginsburgi* and *M. wenzensis* from Europe (*Stach, 1959*; *Wolsan, 1989a*; *Anderson, 1994*; *Sato et al., 2003*; *Montoya, Morales & Abella, 2011*). Each of these molecular estimates for the divergence of *Gulo* from other closely related gulonines (*Martes* and *Pekania*) are long after the earliest records of *Plesiogulo*, which date to the middle Miocene (MN6) of Turkey and Kazakhstan, between 15.2 and 12.5 Ma (*Schmidt-Kittler, 1976*; *Montoya, Morales & Abella, 2011*). Though it seems clear that *Plesiogulo* falls along a different lineage, new and/or older records could clarify the origin of *Gulo*.

Here, we describe a new species of *Gulo* from the Early Pliocene age Gray Fossil Site of Tennessee (Fig. 1). The specimen described here represents the earliest record of *Gulo* and predates other records of the genus by over one million years. Similarity of this new species to early fishers (*Pekania*), suggests that wolverines may have originated in North America. In light of this discovery and other fossil records of *Gulo*, we also discuss the historical biogeography and ecology of wolverines, as well as their commonly accepted 'cold-adapted' nature.

## MATERIALS AND METHODS

The electronic version of this article in Portable Document Format (PDF) will represent a published work according to the International Commission on Zoological Nomenclature (ICZN), and hence the new names contained in the electronic version are effectively published under that Code from the electronic edition alone. This published work and the nomenclatural acts it contains have been registered in ZooBank, the online registration system for the ICZN. The ZooBank LSIDs (Life Science Identifiers) can be resolved and the associated information viewed through any standard web browser by appending the LSID to the prefix http://zoobank.org/. The LSID for this publication is: [29DF929D-D054-4912-A2B1-FFEEFD4BDE1B]. The online version of this work is archived and available from the following digital repositories: PeerJ, PubMed Central and CLOCKSS.

Dental nomenclature follows *Ginsburg (1999)*. Measurements of the teeth, to the nearest 0.01 mm, were made using Mitutoyo Absolute digital calipers. Measurements include anteroposterior length and transverse breadth of the teeth. For the P4, maximum transverse breadth was measured at both the protocone ($P4W_{pro}$) and the metastyle ($P4W_{met}$), and for the M1 lengths were measured for both inner ($M1L_{int}$) and outer lobes ($M1L_{ext}$).

Measurements were taken, and comparisons were made, with specimens of extant and extinct mustelids from several collections. Extant samples included *Gulo gulo* ($n = 36$), *Pekania pennanti* ($n = 22$), *Martes americana* ($n = 11$), *Martes flavigula* ($n = 1$), *Martes foina* ($n = 1$), *Martes martes* ($n = 1$), *Eira barbara* ($n = 5$); this includes all extant gulonine species other than *Martes melampus* and *Martes zibellina*. Fossil samples included Pleistocene and Holocene specimens of *Gulo* and *Pekania* from North America, late Miocene specimens of *Pekania occulta* and *Plesiogulo marshalli* from North America, as well as casts of *Pekania palaeosinensis* from Asia and *Plesiogulo* from North America. All material was also compared to specimens and measurements in a wide range of publications (including *Zdansky, 1924*; *Gidley & Gazin, 1933*; *Gidley & Gazin, 1938*; *Chardin & Leroy, 1945*; *Stach, 1959*; *Anderson, 1970*; *Anderson, 1998*; *Kurtén, 1970*; *Kurtén, 1973*; *Hendey, 1978*; *Harrison, 1981*; *Bryant, 1987*; *Wolsan, 1989a*; *Wolsan, 1989b*; *Ginsburg & Morales, 1992*; *Döppes, 2001*; *Haile-Selassie, Hlusko & Howell, 2004*; *Montoya, Morales & Abella, 2011*; *Peigné, 2012*; *Wang, Tseng & Takeuchi, 2012*). Complete measurement data for all mustelids studied are included in Table S2.

Fossil and modern specimens used for comparative illustrations were photographed with the alveolar margin of the upper carnassial (P4) parallel to the photographic plane. This is important to note, as most published fossil mustelids have been photographed with

the palate parallel to the photographic plane (Fig. S1A); in that orientation the alveolar margins of the cheek teeth are at a slightly oblique angle, with the lateral portions slightly ventral to the medial. A consequence of this difference in photography and illustration methodology is that some of the lingual portions of the tooth, like the P4 lingual cingulum, are not obscured using our methodology (Fig. S1B).

Specimen Repositories—ETMNH, East Tennessee State University Museum of Natural History—Fossil Collection, Gray, Tennessee; ETVP, East Tennessee State University Museum of Natural History—Comparative Collection, Johnson City, Tennessee; LACM, Natural History Museum of Los Angeles County, Los Angeles, California; MVZ, Museum of Vertebrate Zoology, University of California, Berkeley, California; UCLA, Donald R. Dickey Collection of the University of California, Los Angeles, Los Angeles, California; USNM, United States National Museum of Natural History (Smithsonian Institution), Washington, D.C.

# GEOLOGICAL SETTING

The Gray Fossil Site in northeastern Tennessee includes deposits that represent an ancient sinkhole containing a small, but deep lake, which gradually filled with sediment (*Shunk, Driese & Clark, 2006*; *Shunk, Driese & Dunbar, 2009*). Sediment cores have revealed a series of rhythmites in the upper lacustrine strata, which alternate between fine-grained silty clay layers and coarse-grained, organic rich layers (*Shunk, Driese & Clark, 2006*; *Shunk, Driese & Dunbar, 2009*). *Shunk, Driese & Dunbar (2009)* estimated the sinkhole lake filled with sediment in approximately 4,500 to 11,000 years. Within the sedimentary layers are well-preserved and diverse flora and fauna (ex. *Parmalee et al., 2002*; *Wallace & Wang, 2004*; *Mead et al., 2012*; *Zobaa et al., 2011*; *Ochoa et al., 2012*; *Ochoa et al., 2016*; *Worobiec, Liu & Zavada, 2013*), which indicate a forested environment was present. Macro- and microfossils indicate the presence of a forest dominated by oak (*Quercus*), hickory (*Carya*), and pine (*Pinus*), as well as a variety of herbaceous taxa (*Ochoa et al., 2016*; and references therein). Isotopic analyses of carbon and oxygen from teeth of ungulates at the site indicate a relatively dense forest, with some more open grass-dominated habitats nearby, and climate with little seasonal temperature and precipitation variation (*DeSantis & Wallace, 2008*). *Ochoa et al. (2016)* interpreted the flora at the Gray Fossil Site as indicative of a woodland or woodland savanna environment characterized by frequent disturbance. The occurrence of bald cypress (*Taxodium*) and tupelo (*Nyssa*) leaves and pollen at Gray (*Brandon, 2013*; *Worobiec, Liu & Zavada, 2013*) suggest the presence of humid riparian or wetland areas. Fauna includes many taxa indicative of aquatic environments, such as fish, neotenic salamanders, aquatic turtles, *Alligator*, and beavers (*Parmalee et al., 2002*; *Boardman & Schubert, 2011*; *Mead et al., 2012*; *Jasinski, 2013*; *Bourque & Schubert, 2015*).

Age of the Gray Fossil Site has been previously reported as constrained between 7 and 4.5 Ma, based on the stratigraphic ranges of the rhino *Teleoceras* and ursid *Plionarctos* (*Wallace & Wang, 2004*). *Teleoceras* has commonly been considered an index taxon for the Hemphillian NALMA, but it is worth noting that there are several Pliocene records of *Teleoceras*, including Blancan age specimens from White Bluffs in Washington (*Gustafson,*

*2012*), Beck Ranch in Texas (*Madden & Dalquest, 1990*), and Saw Rock Canyon in Kansas (*Prothero & Manning, 1987*). It is important to note that the record from Beck Ranch in Texas has been interpreted as a tooth reworked from older sediments (*Prothero, 2005*) and the report from Kansas does not refer to any cataloged specimens, but the presence of *Ogmodontomys sawrockensis* indicates Saw Rock Canyon is Blancan in age (*Martin & Peláez-Campomanes, 2014*; *Martin, Peláez-Campomanes & Viriot, 2017*). Some of the latest rhino records in North America may be younger than 4.5 Ma and potentially as young as 3.5 Ma (*Gustafson, 2012*). *Gustafson (2012)* asserted that such records indicate that the presence of *Teleoceras* at a site is insufficient justification for assigning a Hemphillian age to the fauna, as has been done for a variety of eastern U.S. localities, specifically the Palmetto fauna (*Webb et al., 2008*) and Pipe Creek Sinkhole (*Farlow et al., 2001*; though see (*Martin, Goodwin & Farlow, 2002*; *Martin, 2010*) for a revised age of that site).

A number of recently identified taxa from the Gray Fossil Site have good fossil records and limited stratigraphic ranges. Based on first and last appearance data (FAD, LAD) of these taxa derived from the MIOMAP/FAUNMAP Databases (*Carrasco et al., 2007*; *Graham & Lundeliu Jr, 2010*; http://www.ucmp.berkeley.edu/neomap/), NOW Database (*Fortelius, 2013*; http://pantodon.science.helsinki.fi/now/), and recent publications, we can produce a revised estimate of the age of the Gray Fossil Site. The cricetids *Neotoma* and *Symmetrodontomys* have their first appearances near the Hemphillian—Blancan transition, approximately 4.9 or 4.8 Ma (*Martin, 2000*; *Martin, Goodwin & Farlow, 2002*; *Bell et al., 2004*; *Lindsay, 2008*; *Martin, Peláez-Campomanes & Viriot, 2017*). The leporid *Notolagus lepusculus* and the mephitid *Buisnictis breviramus* appeared in, and are restricted to, the early Blancan (FAD 4.9 Ma). Among other taxa at the site are the cricetid *Repomys* (FAD 6.6 Ma) and the leporid *Alilepus vagus* (FAD 5.9), which appeared in the Hemphillian and survived into the Blancan. The dromomerycid *Pediomeryx* appeared earlier in the Miocene, but disappeared near the Hemphillian—Blancan transition (LAD 4.7 Ma) (*Voorhies, 1990*; *Janis & Manning, 1998*). However, a recently described record of *Pediomeryx* from the Lee Creek Mine Local Fauna in North Carolina (*Eshelman & Whitmore Jr, 2008*) is early Blancan in age, as *Marx & Fordyce (2015)* bracket the age of the lower Yorktown Formation (including Lee Creek Mine) between 4.9 and 3.9 Ma.

In sum, none of the mammal genera at the Gray Fossil Site are restricted to the late Miocene or Hemphillian NALMA, and some taxa are actually characteristic of early Blancan (Early Pliocene) faunas. For example, the Gray Fossil Site and the early Blancan age Beck Ranch fauna in Texas (*Dalquest, 1978*) share the following (*Teleoceras*, *Neotoma*, *Symmetrodontomys*, *Notolagus lepusculus*, *Buisnictis breviramus*). Other than *Neotoma*, none of these taxa survived later than the early Blancan (~2.5 Ma; *Bell et al., 2004*). Considering the Gray Fossil Site fauna in total, a revised estimate of the age of the Gray Fossil Site is Early Pliocene near the Hemphillian–Blancan transition, likely between 4.9 and 4.5 Ma (Fig. 2).

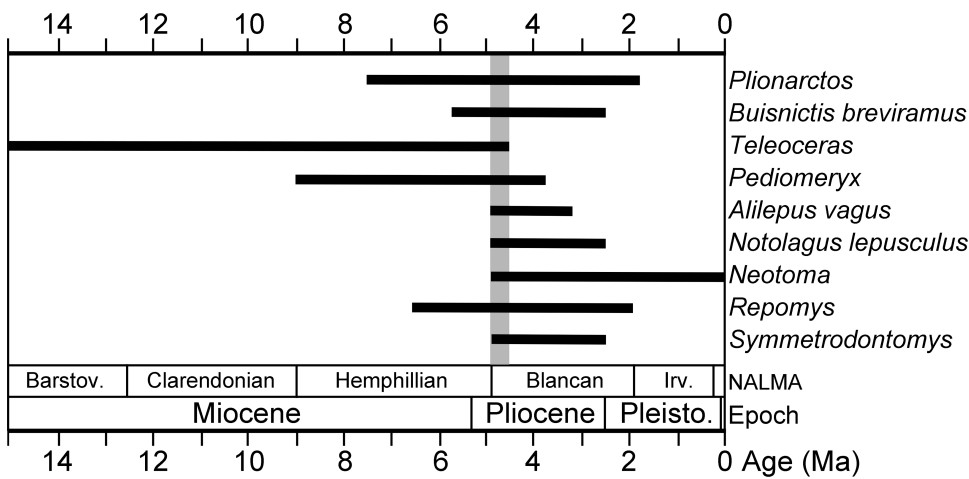

**Figure 2 Stratigraphic ranges of selected mammals from the Gray Fossil Site in Tennessee.** The black bars indicate stratigraphic ranges of genera and species based on first and last appearance dates (data sources are listed in the Geologic Setting section). Overlap in ranges of taxa, between 4.9 and 4.5 Ma, is highlighted with a gray bar.

# RESULTS

## Systematic paleontology

Class MAMMALIA *Linnaeus, 1758*
Order CARNIVORA *Bowdich, 1821*
Family MUSTELIDAE *Fischer von Waldheim, 1817*
Subfamily GULONINAE *Gray, 1825*
Genus *Gulo Pallas, 1780*

**Type Species**—*Gulo gulo*.
**Included Species**—*Gulo minor*, *Gulo schlosseri*, *Gulo sudorus* (new species).
**Distribution**—Early Pliocene of Tennessee, middle Pliocene of Russia, Pleistocene to recent of North America and Eurasia (see Table S1 for a listing of fossil sites).

*Gulo sudorus* New Species
(Fig. 3, Table 1)

**Holotype**—ETMNH 3663, partial right maxilla with P2 and P4.
**Locality**—Gray Fossil Site, Washington County, Tennessee.
**Age**—Early Pliocene (earliest Blancan).
**Diagnosis**—P2 broad and robust, and P4 enlarged with a broad paracone and metastyle; these features are typical of *Gulo*, but not observed in any other gulonines. P4 protocone anteriorly positioned and infraorbital foramen is oval, unlike *Eira*. P4 lacks an external median rootlet, unlike *Pekania*. P4 protocone and parastyle larger than in extant *Gulo gulo*

Samuels et al. (2018), *PeerJ*, DOI 10.7717/peerj.4648

**Table 1 Dental measurements (in mm) of gulonines and some other mustelid species.** Measurements derived from literature sources include the following: *Gulo gulo* (Holocene) from *Anderson (1998)*, *Gulo gulo* (Pleistocene) from *Döppes (2001)* and *Baryshnikov (2015)*, *Gulo schlosseri* from *Gidley & Gazin (1933)* ("*Gulo gidleyi*") and *Bonifay (1971)*, *Iberictis azanzae* from *Ginsburg & Morales, 1992*, *Plesiogulo brachygnathus* from *Zdansky (1924)*, *P. crassa* and *P. praecocidens* from *Kurtén (1970)*, *P. lindsayi* and *P. marshalli* from *Harrison (1981)*, *P. monspessulanus* from *Hendey (1978)*, *P. botori* from *Haile-Selassie, Hlusko & Howell (2004)*, *Sminthosinis bowleri* from *Bjork (1970)*. *Pekania diluviana* from *Gidley & Gazin (1933)*, *P. palaeosinensis* from *Zdansky (1924)* and *Wang, Tseng & Takeuchi (2012)*, *Ischyrictis zibethoides* from *Peigné (2012)*, *Martes ginsburgi* from *Montoya, Morales & Abella (2011)*, and *M. wenzensis* from *Stach (1959)*. Complete listing of measurements for individuals are included in Table S2. A dagger symbol (†) is used to represent extinct species.

| Species | Specimen No. | Locality | P2L | P2W | P4L | P4W$_{pro}$ | P4W$_{met}$ | M1L$_{ext}$ | M1L$_{int}$ | M1W | P2L/P4L | P4W$_{pro}$/P4L | P4W$_{met}$/P4L | P4L/M1L$_{int}$ |
|---|---|---|---|---|---|---|---|---|---|---|---|---|---|---|
| *Gulo sudorus* new species | ETMNH 3663 | Gray Fossil Site, TN | 7.93 | 6.05 | 21.23 | 13.19 | 8.09 | | | | 0.374 | 0.621 | 0.381 | |
| *Gulo gulo* (Recent) | Mean (*n* = 36) | Various | 6.49 | 4.38 | 19.79 | 11.49 | 7.28 | 7.10 | 8.02 | 13.89 | 0.328 | 0.581 | 0.385 | 2.646 |
| | Min. | | 5.40 | 3.70 | 17.70 | 10.30 | 6.30 | 6.16 | 7.29 | 12.88 | 0.303 | 0.518 | 0.359 | 2.511 |
| | Max. | | 7.80 | 5.29 | 23.90 | 14.04 | 9.30 | 8.12 | 8.77 | 14.92 | 0.402 | 0.619 | 0.409 | 2.769 |
| *Gulo gulo* (Holocene) | Min. (*n* = 5) | Middle Butte Cave, ID | 6.9 | 3.9 | 18.0 | 9.6 | | | 7 | 11.8 | 0.317 | 0.522 | | 2.552 |
| | Max. | Moonshiner Cave, ID | 5.7 | 4.7 | 21.5 | 12.1 | | | 8.1 | 13.8 | 0.321 | 0.581 | | 2.727 |
| | Min. (*n* = 14) | | 5.8 | 3.6 | 18.0 | 9.4 | | | 6.6 | 11.9 | 0.315 | 0.533 | | 2.571 |
| | Max. | | 7.0 | 5.0 | 22.2 | 12.9 | | | 8.7 | 14.2 | 0.322 | 0.563 | | 2.654 |
| *Gulo gulo* (Pleistocene–Eurasia) | Mean (*n* = 18) | Various | 7.17 | 4.82 | 22.02 | 12.96 | | | | | 0.324 | 0.590 | | |
| | Min. | | 6.6 | 4.3 | 19.7 | 11.5 | | | | | 0.301 | 0.550 | | |
| | Max. | | 7.9 | 5.5 | 23.2 | 14.4 | | | | | 0.364 | 0.619 | | |
| *Gulo schlosseri* (Pleistocene–North America) | USNM V8175 | Cumberland Cave, MD | 7.38 | 4.9 | 18.9 | 11 | 7.3 | 6.98 | 9.18 | 14.46 | 0.338 | 0.582 | 0.386 | 2.379 |
| | USNM V8176 | | | | 21.84 | 11.65 | 8.5 | | | | | 0.533 | 0.389 | |
| *Pekania occulta*† | JODA 15214 | Rattlesnake Fm., OR | | 3.0 | 13.25 | 8.78 | 5.31 | 5.56 | 8.54 | 12.84 | | 0.663 | 0.401 | 1.552 |
| *Pekania palaeosinensis*† | IVPP V 18408 | Baogeda Ula, China | 5.8 | 2.19 | 12.38 | 6.29 | 3.95 | 5.7 | 6.5 | 10.4 | 0.468 | 0.508 | 0.319 | 1.692 |
| | Mean (*n* = 7) | Baode, China | 5.12 | 2.32 | 10.54 | 6.07 | 4 | | | | 0.486 | 0.576 | 0.364 | |
| | Min. | | 4.2 | 1.9 | 9.0 | 4.6 | | | | | 0.467 | 0.511 | | |
| | Max. | | 5.7 | 2.5 | 11.5 | 6.7 | | | | | 0.496 | 0.583 | | |
| *Pekania diluviana*† | USNM V8010 | Cumberland Cave, MD | | | 10.2 | 6 | 3.9 | 5 | 6 | 9.5 | | 0.588 | 0.382 | 1.7 |
| *Pekania pennanti* | Mean (*n* = 22) | Various | 5.45 | 2.79 | 11.54 | 6.98 | 4.22 | 5.78 | 6.86 | 10.09 | 0.472 | 0.605 | 0.364 | 1.695 |
| | Min. | | 4.66 | 2.04 | 9.98 | 5.83 | 3.34 | 4.87 | 5.47 | 8.43 | 0.423 | 0.575 | 0.335 | 1.521 |
| | Max. | | 6.42 | 3.18 | 13.36 | 7.88 | 4.91 | 6.52 | 7.77 | 11.51 | 0.513 | 0.650 | 0.403 | 1.969 |
| *Martes ginsburgi*† | VV-11759 | Venta del Moro, Spain | | | 8.9 | 5.7 | | | 5.4 | 8.3 | | 0.640 | | 1.648 |
| *Martes wenzensis*† | Holotype | Węże 1, Poland | 5.8 | 3.0 | 12.0 | | 5.2 | | 6.6 | 10.5 | 0.483 | | 0.433 | 1.818 |
| *Martes americana* | Mean (*n* = 11) | Various | 4.33 | 2.00 | 7.80 | 4.86 | 2.84 | 3.79 | 5.23 | 7.88 | 0.557 | 0.622 | 0.362 | 1.536 |
| | Min. | | 3.21 | 1.57 | 6.50 | 3.90 | 1.90 | 3.09 | 3.78 | 6.01 | 0.473 | 0.582 | 0.292 | 1.334 |
| | Max. | | 5.15 | 2.23 | 9.17 | 5.80 | 3.51 | 4.68 | 6.64 | 9.36 | 0.625 | 0.642 | 0.396 | 1.728 |

**Table 1** (*continued*)

| Species | Specimen No. | Locality | P2L | P2W | P4L | P4W$_{pro}$ | P4W$_{met}$ | M1L$_{ext}$ | M1L$_{int}$ | M1W | P2L/P4L | P4W$_{pro}$/P4L | P4W$_{met}$/P4L | P4L/M1L$_{int}$ |
|---|---|---|---|---|---|---|---|---|---|---|---|---|---|---|
| *M. martes* | LACM 74508 | | 5.34 | 2.78 | 9.25 | 6.12 | 3.36 | 4.72 | 7.81 | 9.55 | 0.577 | 0.662 | 0.363 | 1.184 |
| *M. flavigula* | LACM 8229 | | 5.22 | 2.6 | 10.11 | 6.42 | 3.48 | 4.02 | 5.31 | 9.57 | 0.516 | 0.635 | 0.344 | 1.904 |
| *M. foina* | ETVP 5535 | | 4.1 | 2.43 | 9.37 | 5.83 | 3.36 | 5.2 | 5.55 | 9.22 | 0.438 | 0.622 | 0.359 | 1.688 |
| *Eira barbara* | Mean ($n=5$) | Various | 3.85 | 2.84 | 10.06 | 6.90 | 3.66 | 3.60 | 4.68 | 8.73 | 0.382 | 0.687 | 0.365 | 2.183 |
| | Min. | | 3.30 | 2.56 | 9.23 | 6.47 | 3.58 | 2.65 | 3.61 | 8.17 | 0.347 | 0.610 | 0.337 | 1.905 |
| | Max. | | 4.31 | 3.04 | 10.62 | 7.49 | 3.85 | 4.25 | 5.57 | 9.52 | 0.406 | 0.743 | 0.417 | 2.637 |
| *Sminthosinis bowleri*† | UMMP 52868 | Hagerman, ID | 5.38 | 2.15 | 9.6 | 5.35 | | | 4.76 | 7.47 | 0.560 | 0.557 | | 2.017 |
| | UMMP 55214 | | 4.98 | | 9.65 | 5 | | | | | 0.516 | 0.518 | | |
| *Ischyrictis zibethoides*† | Mean ($n=4$) | Various, Europe | 8.5 | 4.6 | 15.95 | 10.1 | 5.6 | 7.03 | 8.58 | 15.25 | 0.478 | 0.630 | 0.329 | 1.964 |
| | Min. | | | | 14.0 | 7.9 | 4.6 | 6.65 | 6.85 | 13.75 | | 0.564 | 0.4 | 1.798 |
| | Max. | | | | 17.8 | 12 | 6.6 | 7.4 | 9.9 | 16.5 | | 0.674 | | 2.263 |
| *Plesiogulo brachygnathus*† | Mean ($n=10$) | Shansi, China | 8.15 | 5.85 | 19.24 | 12.87 | | 9.65 | 13.54 | 16.88 | 0.420 | 0.669 | | 1.428 |
| | Min. | | 7.6 | 5.0 | 17.1 | 11.1 | | 8.4 | 12 | 13.8 | 0.393 | 0.649 | | 1.258 |
| | Max. | | 8.9 | 6.7 | 20.5 | 14.0 | | 11.7 | 16.3 | 18.4 | 0.445 | 0.683 | | 1.583 |
| *Plesiogulo crassa*† | Mean | Paote, China | 8.15 | 5.9 | 19.48 | 12.92 | | 8.4 | 13.22 | 16.98 | 0.418 | 0.663 | | 1.473 |
| | Min. | | 7.8 | | 18.3 | 12.3 | | 7.6 | 11.9 | 15.8 | | | | |
| | Max. | | 8.3 | | 20.8 | 14.2 | | 9 | 14.8 | 17.8 | | | | |
| *Plesiogulo minor*† | Holotype | K'ingyang, China | | | 17 | 10.5 | | 7.9 | 11.2 | 14.3 | | 0.618 | | 1.518 |
| *Plesiogulo praecocidens*† | UPI No. M19 (5) | Paote, China | | | 17.2 | 10.9 | | 7.8 | 12.4 | 13.8 | | 0.634 | | 1.387 |
| *Plesiogulo lindsayi*† | F:AM 49384 | Wikieup, Arizona | 9.5 | 6.9 | 23.5 | 17.3 | | 9.7 | 14.6 | 20.6 | 0.404 | 0.736 | | 1.610 |
| *Plesiogulo marshalli*† | Mean ($n=8$) | Various, North America | 8.01 | 5.83 | 19.75 | 13.25 | 8.59 | 9.88 | 13.95 | 17.58 | 0.416 | 0.670 | 0.434 | 1.411 |
| | Min. | | 7.84 | 5.7 | 18.2 | 12.1 | 7.81 | 8.3 | 12.45 | 17.0 | | 0.640 | 0.422 | 1.238 |
| | Max. | | 8.2 | 5.9 | 22.04 | 15.0 | 9.80 | 11.87 | 15.9 | 18.5 | | 0.696 | 0.445 | 1.749 |
| *Plesiogulo monspessulanus*† | L40042 | Langebaanweg, South Africa | 9.7 | 7.2 | 23.2 | 15.6 | | | 15.4 | 18.6 | 0.418 | 0.672 | | 1.506 |
| *Plesiogulo botori*† | KNM-NK 41420 | Narok, Kenya | | | 24.5 | 16.7 | | 10.1 | 15.9 | 21.2 | | 0.682 | | 1.541 |
| *Iberictis azanzae*† | Holotype | Artesilla, Spain | | | 15.5 | ?9.7 | | | 10.9 | ?16 | | 0.626 | | 1.422 |

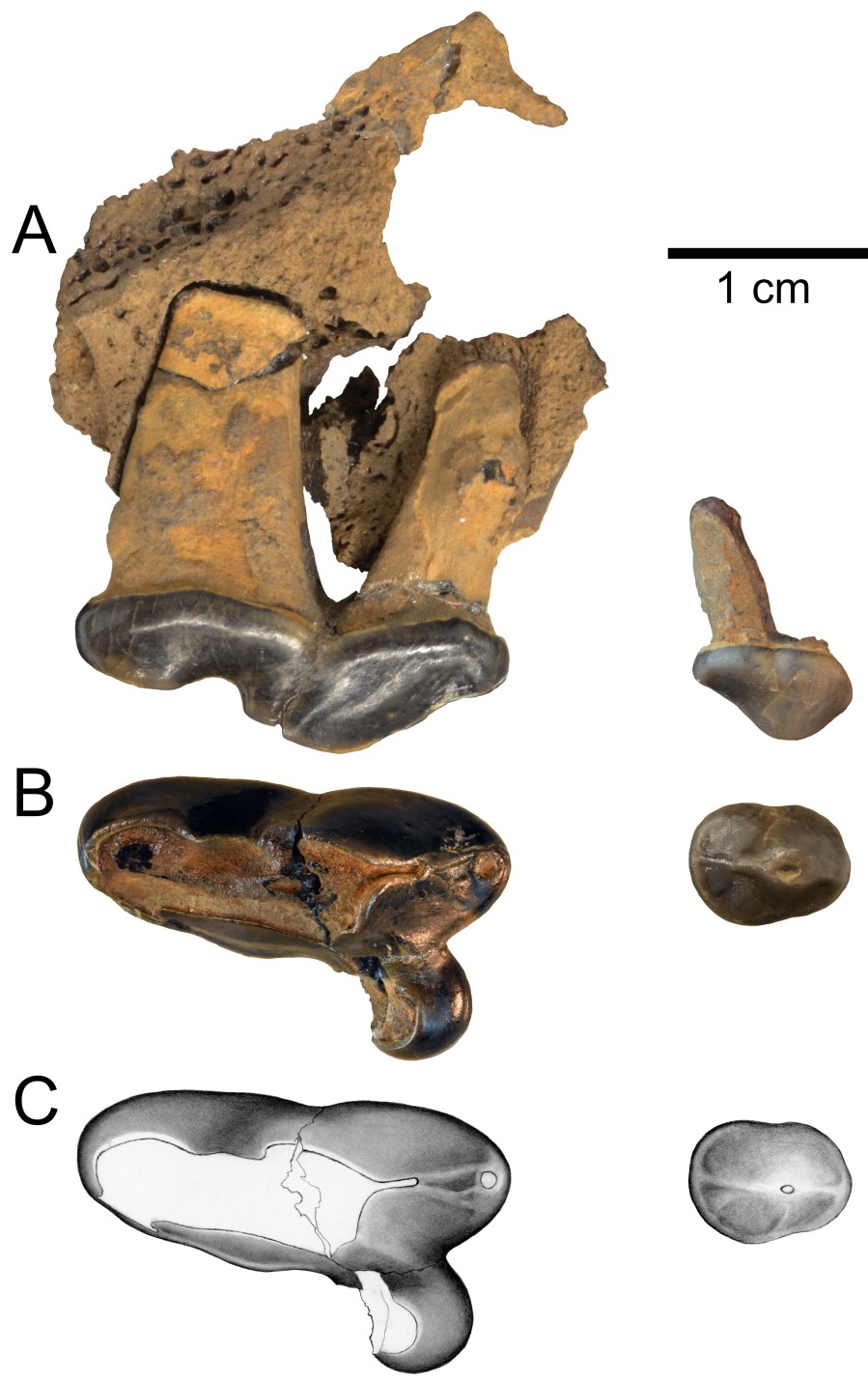

**Figure 3** **Holotype of *Gulo sudorus* (ETMNH 3663) from the Gray Fossil Site, Tennessee.** Specimen consists of a right P2 and maxilla fragment with P4. (A) Lateral view. (B) Occlusal view. (C) Original illustration of the specimen by Keila Bredehoeft. Scale bar equals 1 cm. Photographs by Joshua Samuels.

and all other previously described species of *Gulo*. P4 parastyle includes a small, conical parastylar cusp. P4 metastyle tapers distally, unlike the posteriorly broad and squared metastyle of other *Gulo* species. P2 not as strongly reduced as in other members of the genus. Infraorbital foramen is located above the anterior root of the P4, rather than anterior to the P4 as is typical of other species of *Gulo*.

**Etymology**—"Sudorus" from the Latin for "sweaty". In reference to the warm, humid climate present in the Early Pliocene of Tennessee relative to the typical boreal habitat that the modern taxon is known for inhabiting.

**Description**—The holotype specimen (ETMNH 3663) consists of a left maxilla fragment with the P2 and P4 (Fig. 3). There is little of the maxilla preserved, though some anatomical information can be derived from the specimen. A portion of the infraorbital foramen is preserved above the anterior root of the P4 and its lateral margin suggests an oval shape; the foramen is approximately 10.8 mm in dorsoventral diameter. The P2 crown is intact and only a small portion of the posterior surface of the P4 protocone is missing. The P2 shows minor, and P4 moderate, wear; indicating that this was an adult individual.

The P2 is double-rooted, robust, and distinctly larger than in other members of the genus (Table 1). Though it is larger in absolute terms, the size of the P2 relative to the P4 in ETMNH 3663 falls within the range of variation observed in extant *Gulo gulo* (P2L/P4L, Table 1). The P2 has a single principal cusp and a ridge running along its midline, most distinct posterior to the principal cusp. There is a low, relatively indistinct cingulum on the P2, which is most prominent along its posterior margin.

The P4 is robust and three-rooted, as is typical of the genus. Also typical of *Gulo*, there is no external median rootlet present on the P4, which is a feature that has been used to characterize *Pekania* (though see comparisons below). The tooth bears a distinct parastyle and a relatively larger protocone than in other species of *Gulo*. The parastyle bears a small, round parastylar cusp. A pair of low ridges run along the anterior surface of the paracone; the lateral ridge ends at the posterobuccal edge of the parastylar cusp, the medial ridge forks in two and ends just posterior to the parastyle. Like other gulonines, *Gulo sudorus* has a relatively deep inflection of the anterior portion of the P4, between the parastyle and protocone. The protocone is particularly large and extends anteromedially from the base of the paracone, projecting nearly as far anteriorly as the parastyle. The metastyle of *G. sudorus* is mesially broad, tapers distally, and bears a distinct lingual cingulum. There is a small cingulum along the base of the metastyle on the lingual surface of the tooth and a subtle remnant of an anterior cingulum along the anterior margin of the protocone, both of which are common in known gulonines.

**Comparisons**—Overall, the morphology of ETMNH 3663 is similar to known fossil and modern specimens of *Gulo* and other gulonine mustelids (Fig. 4), but has a set of features that distinguish it from other taxa. As in extant and Pleistocene specimens of *Gulo*, the infraorbital foramen of ETMNH 3663 has an oval shape, but it is located more posteriorly, above the anterior root of the P4, rather than above the P3 or anterior margin of the P4 as in *G. gulo* (Harrison, 1981). Position and shape of the infraorbital foramen in extant and fossil *Pekania* and *Martes* are variable, either oval or round and anterior to the P4 as in *Gulo* or above the anterior root of the P4. In *Eira* the infraorbital foramen is round and

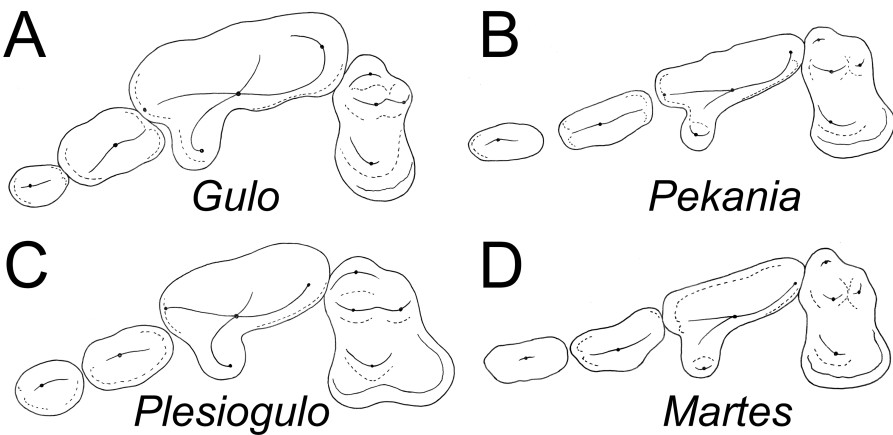

**Figure 4** **Occlusal morphology of gulonine mustelids and _Plesiogulo_.** (A) _Gulo gulo_ (ETVP 291). (B) _Pekania pennanti_ (ETVP 601). (C) _Plesiogulo marshalli_ (composite of FAM 23386 and 49230). (D) _Martes americana_ (NAUQSP 2015). Images are scaled to the same P2–M1 length. Original illustrations by Keila Bredehoeft.

located above the anterior root or middle of the P4. In _Plesiogulo_ the infraorbital foramen is relatively round in cross section and located above the anterior root or middle of the P4 (_Harrison, 1981_).

There is a deep inflection between the parastyle and protocone of the P4 in ETMNH 3663 and _G. gulo_, and in the latter that inflection is occupied by the posterior margin of the P3 (Fig. 4A). In extant and fossil gulonines, a similar arrangement is also seen in individuals of _Martes flavigula_, _M. martes_, and _Pekania palaeosinensis_, but not in _Eira barbara_, _M. americana_, _P. occulta_, _P. diluviana_, or _P. pennanti_.

While a parastylar cusp is occasionally present in the P4 of _Gulo gulo_, it is never as substantial as in ETMNH 3663. Similarly, the protocone is proportionately larger than in extant _Gulo_ and other fossils of the genus (Table 1). Presence of a large protocone and parastyle is typical of other gulonine mustelids, like _Pekania_, _Martes_, and _Eira_; though the protocone is more posteriorly positioned in _Eira_ than other gulonines. A distally tapered P4 metastyle like that of ETMNH 3663 is seen in _Eira_, _Pekania_, and _Martes_, but not in other species of _Gulo_. Specifically, in both _Gulo gulo_ and _G. schlosseri_ the P4 metastyle is broad and squared distally, and bears a distinct u-shaped or square extension at the posterior margin of the metastyle blade. That extension runs anteroposteriorly along the buccal margin of the metastyle, then transversely to join the metastyle blade at a nearly right angle (Fig. 4A).

_Pekania occulta_ from the late Miocene of Oregon has a more robust P4 than other _Pekania_ and _Martes_ species, as well as a particularly large P4 protocone and distally tapered metastyle, and an oval-shaped infraorbital foramen located above the anterior root of the P4. Each of those features are similar to ETMH 3663, though _P. occulta_ is smaller and lacks the robust anterior premolars seen in _Gulo sudorus_. Additionally, as is typical of the genus _Pekania_, _P. occulta_ bears an external median rootlet on the P4, which is not present in ETMNH 3663.

Due to the fragmentary material known for both taxa, the morphology of *Gulo sudorus* and *G. minor* from the middle Pliocene of Asia cannot be directly compared. However, all measurements of the lower dentition of *G. minor* (*Sotnikova, 1982*) fall outside the range of variation of extant and fossil samples of *Gulo*, between 7 and 16% smaller than the smallest extant wolverines studied. In contrast, all dimensions the upper dentition of *G. sudorus*, other than length and width of the P2, fall within the ranges of *G. gulo* and *G. schlosseri*, and all are larger than the means of extant samples (Table 1). As such, it is unlikely that known specimens of *G. sudorus* and *G. minor* are from the same taxon.

A hypothesis of the evolutionary relationships of gulonine mustelids is presented in Fig. 5. Relationships of extant gulonine taxa are based on molecular phylogenetic studies (*Koepfli et al., 2008*; *Sato et al., 2012*; *Li et al., 2014*); estimated divergence times are based on the protein coding region data set in *Li et al. (2014)*, except the divergence time of Guloninae from other "crown" mustelids (Helictidinae, Ictonychinae, Lutrinae, and Mustelinae) which is based on the multidivtime analysis of *Sato et al. (2012)*. Geologic ages of known gulonine fossils are derived from the MIOMAP/FAUNMAP Databases (*Carrasco et al., 2007*; *Graham & Lundeliu Jr, 2010*; http://www.ucmp.berkeley.edu/neomap/), NOW Database (*Fortelius, 2013*; http://pantodon.science.helsinki.fi/now/), and recent publications; the age of *Gulo sudorus* is based on the biostratigraphic framework presented in Fig. 2. Note that the placements of extinct taxa in the phylogeny are not based on a cladistic analysis, but rather morphological comparisons made here and in a number of earlier studies. The placement of the extinct *Pekania palaeosinensis*, *Eirictis*, and *Sminthosinis* are based on *Wang, Tseng & Takeuchi, 2012*, and the placement of *M. wenzensis* is based on *Anderson, 1994*; *Sato et al., 2003*.

The basal placement of *Gulo sudorus* within the genus is based on the following features: (1) infraorbital foramen is positioned posteriorly in *G. sudorus*, as in *Eira* and some species of *Pekania* and *Martes*, and in contrast to the anterior position of the foramen in *G. schlosseri* and *G. gulo*; (2) P2 larger than in any studied specimens of *G. schlosseri* and *G. gulo*, which have reduced anterior premolars; (3) P4 metastyle tapers posteriorly in *G. sudorus*, as in *Ischyrictis zibethoides*, *Eira*, *Pekania*, and *Martes*, and in contrast to the broad and posteriorly squared metastyle of *G. schlosseri* and *G. gulo*; (4) the P4 protocone and parastyle are particularly large in *G. sudorus*, as in *Ischyrictis zibethoides*, *Pekania*, and *Martes*, and larger than in *G. schlosseri* and *G. gulo*. Each of those features of *G. sudorus* are present in other gulonine genera and thus may represent primitive features for the Guloninae. *Gulo minor* is considered derived relative to *G. sudorus*, as the only differences it shows from *G. schlosseri* and *G. gulo* are the following features (from *Sotnikova, 1982*; *Kalmykov, 2015*): (1) smaller body size; (2) weaker curvature of the tooth row, with the m1 trigonid offset at an oblique angle to the p3 and p4, whereas the p3, p4, and m1 trigonid are all aligned in *G. schlosseri* and *G. gulo*; (3) relatively elongate and narrow p3 and p4 (length/width ratio of premolars higher than in any studied samples of other members of the genus). Given the substantial variability in body size of extant wolverines, with the largest studied samples having dentition 25% larger than the smallest (Table 1), the somewhat smaller size of *G. minor* is not interpreted as a particularly substantial difference. The difference in the curvature of the toothrow is also not particularly distinctive, as the

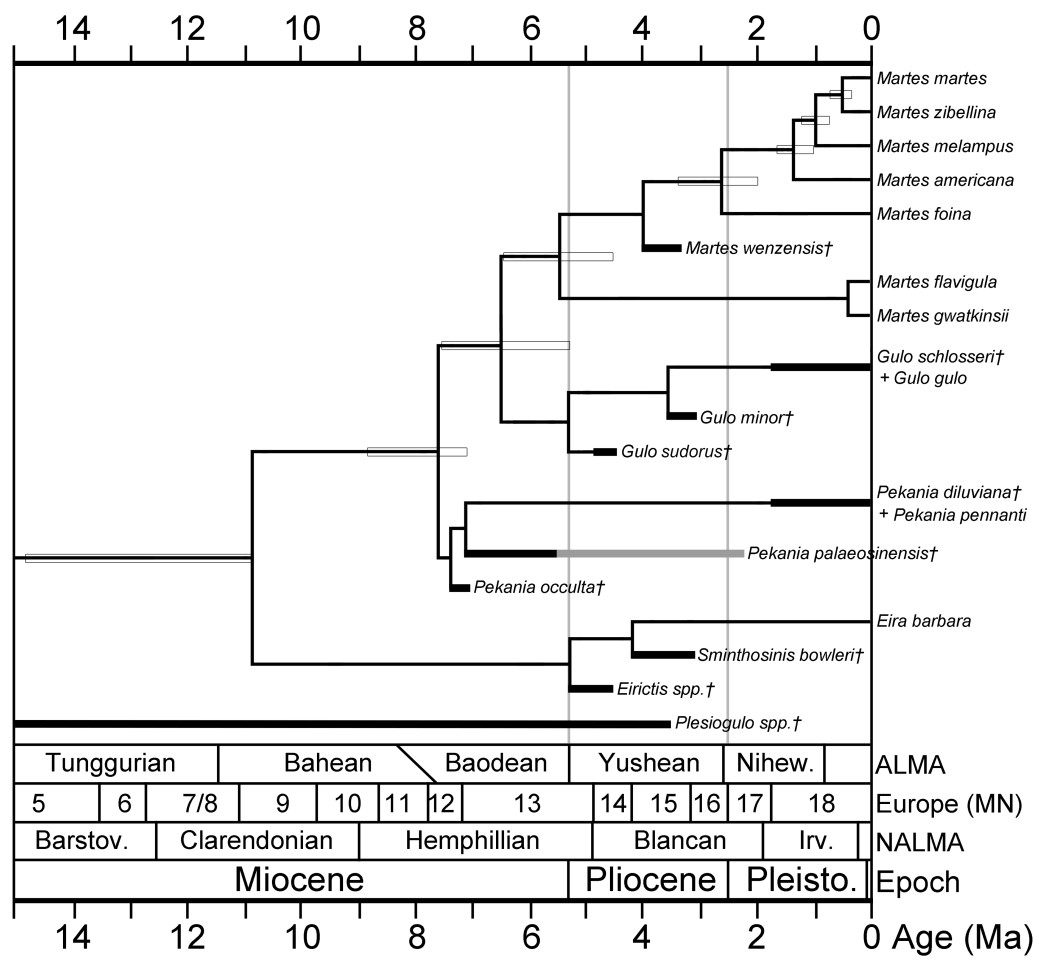

**Figure 5** **Phylogeny of gulonine mustelids with estimated divergence times and geologic ages of known fossils.** Phylogenetic relationships of extant taxa based on cladistics analyses of *Koepfli et al. (2008)*, *Sato et al. (2012)* and *Li et al. (2014)*. Note that the placements of extinct taxa are not based on a cladistic analysis, rather they are based morphological comparisons made in this manuscript and prior studies. Divergence times (indicated by horizontal boxes) are based on protein coding region data set in *Li et al. (2014)*, except divergence time of Guloninae from other "crown" mustelids (Helictidinae, Ictonychinae, Lutrinae, and Mustelinae) based on multidivtime analysis of *Sato et al. (2012)*. Age ranges of fossil taxa (thick black lines) are based on occurrences and references listed in Table S2 and data sources listed in the Geologic Setting section.

dental arcade of extant wolverines is highly variable in terms of rotation and alignment of the anterior premolars (*Jung et al., 2016*). Within the genus *Pekania*, placement of *P. occulta* as the most basal member is based on the following features: (1) robust P4, more robust than in other members of the genus, (2) P4 protocone larger than in other members of the genus. Each of these features are similar to *Gulo*, and thus may represent the primitive state for the *Pekania—Gulo/Martes* clade.

## DISCUSSION

### Origin and evolution of *Gulo*

Origins of the subfamily Guloninae and the genus *Gulo* have long been uncertain, but recently described fossils have substantially improved our understanding of the group's evolution. The earliest species that has been referred to any extant gulonine genus is "*Martes*" *laevidens* from the early Miocene (MN 3) of Germany (*Dehm, 1950*). However, *Sato et al. (2003)* noted the basicranial anatomy of "*M.*" *laevidens* indicated it was not a member of the extant genus *Martes*; the incompletely ossified suprameatal fossa (*Wolsan, 1993*) is a plesiomorphic trait among mustelids (*Hughes, 2012*). There are many other described examples of potential early "*Martes*" species from the early and middle Miocene of Eurasia, North America, and North Africa (*Anderson, 1994*; *Baskin, 1998*; *Ginsburg, 1999*; *Hughes, 2012*); however; most are not closely related to extant gulonine genera, but instead represent stem groups outside of the crown clade Guloninae (*Anderson, 1994*; *Sato et al., 2003*; *Wang, Tseng & Takeuchi, 2012*; *Li et al., 2014*). Moreover, some of these "*Martes*" taxa show similarity to ischyrictines (*Ginsburg & Morales, 1992*; *Montoya, Morales & Abella, 2011*) and, in some cases, have been referred to the ischyrictine genera *Hoplictis*, *Plionictis*, and *Sthenictis* (*Anderson, 1994*; *Baskin, 1998*; *Hughes, 2012*).

Similarities of many of these potential early "*Martes*" species with members of the Guloninae is likely due to the retention of plesiomorphic traits in extant gulonine taxa, or more likely ecomorphological convergence with the genus *Martes*. Complicating the issue, extant gulonines, including species of *Martes* and *Gulo*, are characterized by highly polymorphic dentition (*Wolsan, Ruprecht & Buchalczyk, 1985*; *Wolsan, 1988*; *Wolsan, 1989b*; *Döppes, 2001*; *Jung et al., 2016*), and it seems likely that ancestral forms shared the same level of variation. Given the polymorphism observed in extant taxa and incomplete fossil material known for many early and middle Miocene species, early taxa referred to "*Martes*" are in need of detailed study, which may reveal how they are (or are not) related to gulonines.

Something similar to *Ischyrictis zibethoides*, which is known from the Early and Middle Miocene (MN 5-8) of Europe (*Peigné, 2012*), could be ancestral to gulonines. *I. zibethoides* exhibits a dentition which has some similarity to that of gulonines; the P4 protocone and parastyle are large, and there is a deep inflection in the anterior margin of the tooth. The M1 has a distinctive round lingual margin. Lastly, the age of that taxon is similar to the estimated divergence of gulonines from other mustelids, which molecular estimates placed near 11.0 Ma (*Koepfli et al., 2008*) or 12.65 Ma (*Sato et al., 2012*).

The oldest confirmed gulonines are early records of fishers, including *Pekania occulta* from North America (*Samuels & Cavin, 2013*) and *P. palaeosinensis* from Asia (*Wang, Tseng & Takeuchi, 2012*). *Pekania*, which was formerly considered a subgenus of *Martes*, is distinguished by the presence of an external median rootlet on the upper P4 (*Anderson, 1994*; *Samuels & Cavin, 2013*). Morphology of *P. occulta*, particularly the robust P4 with a large protocone, is similar to extant *Gulo gulo* and *G. sudorus*, suggesting that species may be similar to the shared ancestor of more recent *Pekania* and *Gulo*. The late Miocene ages of *P. occulta* (Hemphillian NALMA, 7.3–7.05 Ma) and *P. palaeosinensis* (Baodean ALMA)

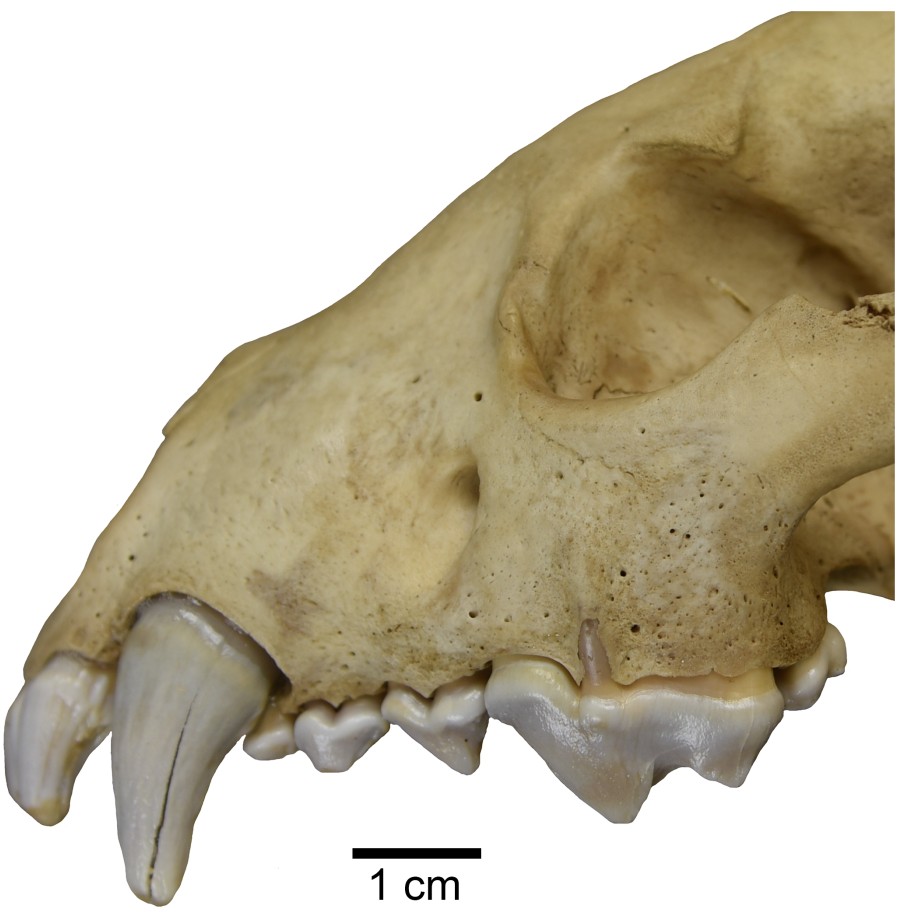

**Figure 6** *Gulo gulo* **(ETVP 269) with an atavistic external median rootlet on the upper P4, which is exposed through the lateral surface of the maxilla.** Scale bar equals 1 cm. Photograph by Joshua Samuels.

are consistent with molecular estimates for divergence of *Pekania* from the *Martes*/*Gulo* clade (*Koepfli et al., 2008*; *Sato et al., 2012*; *Li et al., 2014*).

Interestingly, one studied specimen of recent *Gulo gulo* (ETVP 269) possesses an external median rootlet on both upper carnassials (Fig. 6). Presence of an external median rootlet on the upper P4 is considered diagnostic of the genus *Pekania* (*Anderson, 1970*). Though it is only observed in a single specimen, this apparent atavistic trait lends support to both molecular and fossil evidence indicating a close relationship between *Pekania* and *Gulo*. If *Gulo* evolved from a fisher-like ancestor, as we hypothesize, the occurrence of an external median rootlet in ETVP 269 is not as surprising as it would be if the two taxa were not so closely related.

An early representative of the genus *Martes* may be *M. ginsburgi* (*Montoya, Morales & Abella, 2011*) from the late Miocene of Spain (MN 13), which is similar to both *M. anderssoni* (*Schlosser, 1924*) from latest Miocene/Early Pliocene and *M. zdanskyi* (*Teilhard de Chardin & Leroy, 1945*) from the late Pliocene of China. *M. ginsburgi* lacks an external median rootlet on the P4 and the m1 morphology is distinct from *Pekania* (*Montoya,*

*Morales & Abella, 2011*). *M. ginsburgi* is also very similar to *M. martes*, suggesting that it may be similar to the ancestor of extant martens (*Montoya, Morales & Abella, 2011*). Detailed study of *M. ginsburgi*, *M. anderssoni*, and other late Miocene species referred to the genus should help resolve how these taxa are related to extant martens.

Various authors (*Anderson, 1994*; *Sato et al., 2003*; *Li et al., 2014*) have considered *Martes wenzensis* (*Stach, 1959*) from the Pliocene (MN 15) of Poland (*Wolsan, 1989a*) to be the earliest undoubted member of the genus *Martes*. *M. wenzensis* is known from multiple skulls and characterized by having longer and more robust carnassials than extant *Martes* (*Stach, 1959*). Measurements of *M. wenzensis* fall within the size range of extant and fossil samples of *Pekania* (Table 1), but it clearly lacks the external median rootlet of the P4 in that taxon. Ages of both species, about 7.1–5.3 Ma for *M. ginsburgi* and 4.0–3.3 Ma for *M. wenzensis*, are consistent with molecular divergence estimates of the *Martes/Gulo* clade from other gulonines (*Koepfli et al., 2008*; *Sato et al., 2012*; *Li et al., 2014*); comparable in age to *G. sudorus* and slightly younger than the earliest records of *Pekania*.

There are a number of distinct evolutionary trends that can be recognized by examination of the teeth of gulonines. Compared to other extant gulonine mustelids, *Gulo* has strongly reduced anterior premolars (Table 1, Fig. 4). Both *Bryant (1987)* and *Jung et al. (2016)* have suggested there is a general evolutionary trend in loss/reduction/rotation of the anterior premolars and enlargement of the carnassials in *Gulo*, which is a consequence of the hypercarnivorous diet of wolverines. Since the canines and carnassials are the primary teeth used to kill and process prey, and reduction in snout length increases the bite force of those teeth, it is not surprising that the anterior premolars of *Gulo* are frequently missing or rotated (*Jung et al., 2016*). The P2 of *G. sudorus* is larger than all previously described members of the genus (Table 1). In terms of relative size (P2L/P4L), *G. sudorus* has anterior premolars intermediate in size between the means of *G. gulo* and related taxa like *Pekania* and *Eira*. The premolar morphology of *G. sudorus* suggests that the evolutionary trend of premolar reduction in the genus *Gulo* had already begun in the Early Pliocene.

In addition to reduction of the premolars, *Gulo* has relatively smaller upper molars than other gulonine mustelids (P4L/M1L$_{int}$) (Table 1, Fig. 4; *Tseng et al., 2009*). The P4 length is well over twice as long as the maximum length of the M1 (inner lobe) in all *Gulo* studied; the only other taxa that approach similar proportions are some specimens of *Eira* and the tayra-like *Sminthosinis bowleri* from the Pliocene (early Blancan) of Idaho (*Bjork, 1970*). Similarly, the ratio of P4 length to the transverse width of the M1 was greater than 1.5 in nearly all *Gulo* studied, whereas it was less than 1.25 in all other taxa. As in the premolars, the relative size of the M1 and m1 talonid in *Gulo* likely reflect adaptations for hypercarnivory; reduction of the post-carnassial dentition and shortening the rostrum increase mechanical advantage when biting with both the canines and carnassials.

### *Plesiogulo* and convergent evolution of wolverine-like mustelids

*Plesiogulo*, known from many mid to late Miocene and Pliocene sites in North America, Eurasia, and Africa, has long been discussed as a potential relative of *Gulo* (*Zdansky, 1924*; *Viret, 1939*; *Webb, 1969*; *Kurtén, 1970*; *Hendey, 1978*; *Harrison, 1981*; *Xu & Wei, 1987*; *Alcalá, Montoya & Morales, 1994*; *Pasitschniak-Arts & Larivière, 1995*; *Montoya, Morales*

& Abella, 2011). Careful consideration of the morphology of extant gulonine mustelids (Fig. 4) and recently discovered fossil material suggests that similarity between *Gulo* and *Plesiogulo* is likely a result of convergence on a similar niche, rather than the result of a close relationship. *Eira*, *Pekania*, *Martes*, and *Gulo* all have a distinct P4 parastyle (though it is variably well-developed), but only a distinct anterior cingulum or occasionally a weak parastyle are seen in *Plesiogulo* (contrary to indication that the parastyle is absent; *Harrison, 1981*). The inner lobe of the M1 is expanded posterolingually in *Plesiogulo* to form a broad talon, but not in *Gulo*. Similarly, the m1 talonid is proportionately much longer and wider in *Plesiogulo* than *Gulo*, and has a shallow basin *Stehlin, 1931*; *Teilhard de Chardin & Leroy, 1945*; *Kurtén, 1970*). A number of evolutionary reversals (P4 parastyle reappearance, M1 reduction, M1 talon reduction, m1 talonid reduction) would be required for *Gulo* to have been derived from something like *Plesiogulo*, as has been previously noted (*Zdansky, 1924*; *Webb, 1969*; *Harrison, 1981*; *Xu & Wei, 1987*). On the other hand, few changes, other than increased size and robustness of the carnassial and premolars, are necessary for derivation of *Gulo* from something similar to early species of *Pekania* (*P. occulta*, *P. palaeosinensis*).

Rather than showing similarity to gulonine mustelids, *Plesiogulo* is most similar to the ischyrictine *Iberictis*, which is known from the early Miocene (MN4) of Spain and France (*Ginsburg & Morales, 1992*). Both *Plesiogulo* and *Iberictis* display robust anterior premolars with strong cingula, a P4 with a large protocone and relatively broad metastyle, large M1 with a posterolingually expanded inner lobe, and relatively broad and elongate m1 talonid. In contrast, *Gulo* and *Pekania* have a relatively narrow P4 metastyle and M1 with inner lobe only slightly longer than the outer lobe, and *Gulo* also has a distinctly reduced M1 and m1 talonid. In *Iberictis* and most species of *Plesiogulo*, the m1 has a distinct metaconid that is approximately equal in height to the paraconid, while the m1 of *Gulo* has no distinct metaconid. As in *Gulo*, the m1 metaconid is absent in *Plesiogulo praecocidens* from the late Miocene of Asia and variably absent in the Pliocene *P. monspessulanus* (*Viret, 1939*; *Kurtén, 1970*; *Bonifay, 1971*; *Hendey, 1978*; *Alcalá, Montoya & Morales, 1994*; *Koufos, 2000*). Additionally, *P. monspessulanus* has a single-rooted p2, while the p2 is double-rooted in *Gulo*, further suggesting the two are not closely related (*Hendey, 1978*).

Another feature distinguishing *Gulo* from *Plesiogulo* are the proportions of the teeth (Table 1, Fig. 4). *Plesiogulo* and *Gulo* both have robust premolars, wide relative to their length, but the anterior premolars are smaller in proportion to the carnassial (P2L/P4L) in all samples of *Gulo*. Similarly, the relative size of the M1 (P4L/M1L$_{int}$) in all samples of *Gulo* is relatively smaller than those of described and examined specimens of *Plesiogulo*. Additionally, no reported specimens of *Plesiogulo* display the premolar rotation commonly seen in *Gulo* (*Jung et al., 2016*).

Many of the features that are similar between *Gulo* and *Plesiogulo* are ones that are functionally important and observed in other carnivorans of similar size and ecology. The extant honey badger *Mellivora*, extinct *Eomellivora* (Mellivorinae), and *Megalictis* (Oligobuninae) all display some strong similarity to *Gulo* and *Plesiogulo*, despite having distinct evolutionary histories (*Zdansky, 1924*; *Kurtén, 1970*; *Hendey, 1978*; *Harrison, 1981*; *Valenciano et al., 2016*; *Valenciano et al., 2017*). Superficial similarity of the teeth (robust posterior premolars and carnassials, trenchant talonids, m1 metaconid loss),

skull (short rostrum, broad and robust zygomatic arches, prominent sagittal crest), and jaw (deep mandibular corpus) are seen in each of these clades. Additionally, the enamel microstructures of the posterior premolars and molars of *Gulo*, *Plesiogulo*, *Mellivora*, and *Eomellivora* all display zigzag Hunter-Schreger Bands (*Stefen, 2001*; *Tseng, Wang & Stewart, 2009*). Independent evolution of similar craniodental morphology and enamel microstructures in these taxa is likely attributable to relatively large body size and similar lifestyles, specifically a hypercarnivorous or durophagous diet (*Van Valkenburgh, 1989*; *Stefen, 1997*; *Stefen, 1999*).

Further support for the hypothesized convergence of *Gulo* and *Plesiogulo*, rather than close relationship, comes from a variety of molecular studies. The earliest occurrences of *Plesiogulo* are in the middle Miocene (MN 6, about 15.2–12.5 Ma; *Schmidt-Kittler, 1976*). That age predates some molecular estimates for the divergence of the Guloninae from other mustelids, including the approximately 11.0 Ma estimate of *Koepfli et al. (2008)*, but not the 12.65 Ma estimate of *Sato et al. (2012)*. However, those early occurrences of *Plesiogulo* precede the molecular estimates of divergence of the *Gulo-Martes* clade from other gulonines by nearly five million years. In contrast, the ages of recently described fossil specimens of *Pekania* (*Wang, Tseng & Takeuchi, 2012*; *Samuels & Cavin, 2013*), and the new *Gulo* specimen described here, agree broadly with molecular divergence estimates of those extant genera (*Koepfli et al., 2008*; *Hughes, 2012*; *Sato et al., 2012*; *Li et al., 2014*; *Malyarchuk, Derenko & Denisova, 2015*). In addition, the ages of *Gulo sudorus* and *G. minor* are similar to the latest records of *Plesiogulo* in North America and Asia. Given the timing of these finds, it is possible that *Plesiogulo* was ecologically replaced by *Gulo* in the Pliocene. It is important to note that the hypothesized convergence described here is supported by morphological and geochronologic evidence, but is not based on analysis of taxa within a cladistic framework, and is thus speculative.

## 'Cold-Adaptation' in *Gulo*

Presence of *Gulo*, which is typically considered an inhabitant of boreal habitats (*Kvam, Overskaug & Sorensen, 1988*; *Pasitschniak-Arts & Larivière, 1995*), at a site with many vertebrate taxa characteristic of warm and/or humid forested habitats (*Alligator*, *Heloderma*, *Pristinailurus*, and *Arctomeles*) and a subtropical forest flora is a unique combination among North American biotas. An interesting analog is known from the middle Pliocene age (MN16a) Udunga fauna of the Transbaikal area of Russia (*Sotnikova & Kalmykov, 1991*; *Erbajeva, Alexeeva & Khenzykhenova, 2003*; *Sotnikova, 2006*; *Erbajeva & Alexeeva, 2013*). The Udunga fauna also has forest-adapted taxa like *Parailurus*, *Parameles*, and *Arctomeles* occurring along with *Gulo* (*Sotnikova, 2006*; *Ogino et al., 2008*; *Erbajeva & Alexeeva, 2013*). Another site with *Gulo* outside its current range is the latest Blancan and Irvingtonian (early Pleistocene) age Vallecito Creek Local Fauna of extreme southern California (*Cassiliano, 1999*). Like Gray, the Vallecito Creek fauna includes *Tapirus* and the procyonid, *Nasua*, which has a current and prehistoric range restricted to the southwestern United States, through Mexico and Central America, and into portions of South America. Occurrences and associated faunas of Gray, Udunga, and Vallecito Creek suggest the boreal distribution of extant *Gulo gulo* may not represent the potential range of habitats occupied

by wolverines, at least in the past. Adaptation to cold, boreal habitats within wolverines may have occurred later, in the late Pliocene or Pleistocene, as global climates became substantially colder and those habitats spread across northern Eurasia and North America.

### Historical biogeography

These finds also have implications for understanding of historical biogeography and evolution of fishers and wolverines. *Koepfli et al. (2008)* and *Zigouris et al. (2013)* both estimate Eurasia as the ancestral area for the origin of these taxa, while *Sato et al. (2012)* restricted the area of origin to Asia. The oldest known probable gulonine is *Ischyrictis zibethoides* from the Early/Middle Miocene (MN 5-8) of Europe. Early records of *Pekania* are known from the late Miocene of North America (early Hemphillian) and Asia (Baodean) (*Wang, Tseng & Takeuchi, 2012*; *Samuels & Cavin, 2013*). Similarly, early records of *Gulo* are known from the earliest Pliocene (early Blancan) of North America and middle Pliocene (Yushean) of Asia (*Sotnikova, 1982*; *Sotnikova, 1995*). While there are many species of "*Martes*" described from the Early and Middle Miocene of Eurasia and North America, the most confidently placed early members of *Martes* are from the Pliocene (MN 15) of Europe (*Stach, 1959*; *Anderson, 1994*; *Sato et al., 2003*). Earliest members of the tayra clade appear later than the other clades, with *Eirictis pachygnatha* from the Early Pliocene (Yushean) of Asia (*Teilhard de Chardin & Leroy, 1945*; *Qiu, Deng & Wang, 2004*; *Wang, Tseng & Takeuchi, 2012*) and *Sminthosinis bowleri* from the middle Pliocene (Blancan) of North America.

Early members of multiple clades (fisher, wolverine-marten, and tayra) show similar distributions in the late Miocene and Pliocene, occurring in both Asia and North America. This suggests connection of these regions via the Bering Land Bridge at the time, which facilitated dispersal of carnivorans between Asia and North America (*Qiu, 2003*). Consequently, a hypothesis for intercontinental dispersals of the wolverine clade is: (1) origin of gulonines in Eurasia, (2) dispersal of *Pekania* from Asia to North America in the late Miocene, (3) divergence of *Gulo* from an ancestor similar to *P. occulta* in the earliest Pliocene of North America, (4) dispersal of *Gulo* to Asia in the Pliocene, following the extinction of *Plesiogulo*, and (5) dispersal of *Gulo* to Europe in the Pleistocene.

## CONCLUSIONS

The new material of *Gulo* described here demonstrates the presence of wolverines in the Early Pliocene of North America, more than 1 million years earlier than other known records of the genus. Differences from the extant species and previously described extinct wolverines indicate the presence of a distinct new species, *G. sudorus*. Similarity of the new species to both early fishers (*Pekania*) and later *Gulo* species suggests that this may represent an intermediate form, which evolved in North America and later dispersed to Eurasia. Presence of *Gulo* in the Pliocene of Tennessee, at a site with a variety of floral and faunal elements indicative of warm/humid climates, suggests that the 'cold-adapted' nature of extant *G. gulo* may be a relatively recent phenomenon. Comparisons of *Gulo* and other members of the Guloninae to *Plesiogulo* suggest that the latter is not closely related to gulonines, but instead likely represents convergence on a similar niche to that occupied

by wolverines. The similar timing of the last records of *Plesiogulo* with the appearance of *Gulo* in North America and Asia suggest the former may have been ecologically replaced by the latter in the Early Pliocene.

## ACKNOWLEDGEMENTS

The following curators and collection managers kindly allowed access to specimens in their care: A Nye and B Compton (ETMNH); K Molina (Donald R. Dickey Collection of the University of California, Los Angeles); C Conroy (University of California, Berkeley— Museum of Vertebrate Zoology); J Dines (Museum of Natural History, Los Angeles County); B Akersten and M Thompson (Idaho Museum of Natural History); and M Brett-Surman and L Gordon (National Museum of Natural History). Constructive reviews by Mieczyslaw Wolsan and an anonymous reviewer helped improve this manuscript greatly.

### Funding

Specimen collection at the Gray Fossil Site in Tennessee was partially funded through a National Science Foundation Grant (NSF Grant #0958985) to SC Wallace and BW Schubert. The remainder of the funding for the project was provided by internal funding from the Don Sundquist Center of Excellence in Paleontology at East Tennessee State University. There was no additional external funding received for this study. The funders had no role in study design, data collection and analysis, decision to publish, or preparation of the manuscript.

### Grant Disclosures

The following grant information was disclosed by the authors:
National Science Foundation Grant: #0958985.
Don Sundquist Center of Excellence in Paleontology at East Tennessee State University.

### Competing Interests

The authors declare there are no competing interests.

### Author Contributions

- Joshua X. Samuels and Keila E. Bredehoeft conceived and designed the experiments, performed the experiments, analyzed the data, prepared figures and/or tables, authored or reviewed drafts of the paper, approved the final draft.
- Steven C. Wallace conceived and designed the experiments, performed the experiments, analyzed the data, authored or reviewed drafts of the paper, approved the final draft.

### Data Availability

   The raw data are included in the Tables S1 and S2.

## New Species Registration

The following information was supplied regarding the registration of a newly described species:

Publication LSID:
urn:lsid:zoobank.org:pub:29DF929D-D054-4912-A2B1-FFEEFD4BDE1B;

*Gulo sudorus* sp. nov.:
urn:lsid:zoobank.org:act:C969FE74-0659-474D-8068-E82DCBF21271.

## Supplemental Information

Supplemental information for this article can be found online at http://dx.doi.org/10.7717/peerj.4648#supplemental-information.

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
