# Peer review of "A new species of Gulo from the Early Pliocene Gray Fossil Site (Eastern United States); rethinking the evolution of wolverines"

_PeerJ, doi:10.7717/peerj.4648_

## Round 0.1 · original submission · Minor Revisions

Both reviewers have commented positively on the content of this manuscript and its suitability for PeerJ. One reviewer has provided some minor comments, whereas the other reviewer has requested two more substantial changes. Taken together the manuscript falls between a major and minor revision. The latter mentioned changes should be addressed in a revision, namely a request for clarification on the use of morphological convergence with character mapping and a request for further details on how this new species is distinguished from members of non-Gulo genera.

·

Basic reporting

no comment

Experimental design

no comment

Validity of the findings

no comment

Additional comments

Minor comments:
1) Line 61: 2012, not “2011”.
2) Line 73: “Koufos, 1914”? Should it be Koufos 1982? The latter is included in the References section but nowhere cited.
3) Lines 73–74, 138 and 871: Gidley & Gazin, not “Gazin & Gidley”.
4) Lines 160–161: “Shunk, Driese, & Clark, 2006; Shunk, Driese, & Dunbar, 2009”, not “Shunk, Driese, & Clark, 2006; 2009”.
5) Lines 226 and 578: Bowdich, not “Bowditch”.
6) Lines 227 and 610: Fischer, not “Fischer von Waldheim”.
7) Line 401: “in” is missing between “1.25” and “all other taxa”.
8) Line 482: Kalmykov, not “Kalmykova”.
9) Line 500: Biogeographic analyses of Sato et al. 2012 restrict this area to Asia.
10) The References section lacks Dalquest 1978 (cited in line 217), Eshelman & Whitmore 2008 and Marx & Fordyce 2015 (both cited in line 211), Farlow et al. 2001 (cited in line 195), Gustafson 2012 (cited in lines 184 and 191), Madden & Dahlquest 1990 (cited in line 185), Stehlin 1931 (cited in line 419), Valenciano et al. 2016, 2017 (both cited in line 451), Voorhies 1990 and Janis & Manning 1998 (both cited in line 209), Webb et al. 2008 (cited in line 194) and Wolsan & Sato 2010 (cited in line 88).
11) Burmeister 1850, Eizirik et al. 2010, Gray 1865, Hilzheimer 1936, Kalmykov 2015, Mckelvey et al. 2014, Nadachowski et al. 2011, Orlov 1941, Repenning 1987, Sato et al. 2009, Woodburne 2004 and Yu et al. 2011 are included in the References section but nowhere cited.

Reviewer 2 ·

Basic reporting

no comment

Experimental design

no comment

Validity of the findings

Two points require clarification and additional detail:

1. The interpretation that Gulo and Plesiogulo are morphologically convergent is supported by geochronologic and dental morphological evidence, but no cladistic framework is used. As such, the claim of evolutionary convergence (the concept itself which is defined within a phylogenetic context) is not supported by mapped character distributions over the tree. If no cladistic framework will be used and the authors wish to suggest convergence in those two taxa, I recommend explicitly identifying this interpretation as a speculation.

2. The diagnosis of G. sudorus in the systematic paleontology section provides differentiation between Gulo species, but not between this new species and other non-Gulo gulonines. The differential diagnosis could be improved by citing characters that exclude the new species from membership in other genera.

Please see general comments for specific sections in the manuscript where the two comments above apply.

Additional comments

In this study the authors describe a new fossil species of Gulo, present evidence that early Gulo lived in warmer and wetter environments than indicated by the genus' present range, and suggest that Gulo and Plesiogulo may be convergent in their adaptation for opportunistic hypercarnivory. Overall, the study presents interesting and important new data in understanding wolverine evolution, and also in a re-assessment of the likely geologic age of a very important fossil locality for understanding pre-Pleistocene mammal evolution in North America. As I mentioned in Section 3 ("validity of the findings") above, I recommend making changes to two important aspects of the paper before it is accepted for publication.

Below I include detailed comments on sections of the manuscript:
-Abstract: diagnosis of convergence is contingent upon a cladistic framework, but because no such framework including Gulo and Plesiogulo is presented in this study, I would recommend the authors present this last sentence as a speculation rather than an interpretation supported by phylogenetic evidence.
-L51: Supp. Table 1 is a very useful compilation of Gulo localities.
-L132: any particular reason these two species of Martes were not examined?
-L142: Within Supp. Table 2, the locality name for Plesiogulo crassa ("Paote") should be consistent with the locality for some of the Pekania palaeosinensis specimens ("Baode"). The two names refer to the same locality. "Baode" is the modern Pinyin spelling, whereas "Paote" is the traditional Wade-Giles spelling. Only one system should be used, for consistency.
-L151: Are ETMNH specimens extant and ETVP specimens fossils? Please clarify the distinction between these two collection acronyms.
-L155: should add "(Smithsonian Institution)" after "National Museum of Natural History".
-L196-221: This entire section is a critical reworking of the biostratigraphic evidence supporting a revised estimate for the age range of the Gray Fossil Site. The reasoning and evidence cited herein would be useful for future studies related to the Gray Fossil Site biota as well. Therefore, I recommend the authors include a summary biostratigraphic range chart to accompany this section, to further visualize the evidence for their interpretation of the age of the locality.
-L235: Figure 2 is cited in text before Figure 1. Please consider re-ordering the figures, or cite Fig. 1 in introduction.
-L235 Table 1: please see previous comment regarding the localities "Baode" and "Paote".
-L239: are there characteristics that unite this new species with other Gulo species, and distinguish them from other gulonines? As written, the diagnosis differentiates species of Gulo, but does not address differentiation of the new species from (thus eliminating it from membership in) other genera? From the comparison section below, the lack of the additional median rootlet on P4 may qualify as a diagnostic character between Gulo and Pekania (or more broadly, Pekania with all other gulonines).
-L854 Fig. 4. The placement of the new species described in this study was not based on a cladistic framework, and thus should be indicated as such. The authors should include justification for the grouping of Gulo species shown in this figure; what are the characteristics supporting G. sudorus as the most basal Gulo? Otherwise, it appears as if the phylogenetic relatinoships in this group mainly reflect their chronological order of appearance in the fossil record.

---

## Round 0.2 · accepted · Accept

Thank you for carefully addressing all points raised by the reviewers (which were few). This revised manuscript fully addresses the two main points raised by Reviewer #2, and as such I recommend the paper be accepted.

#